# Genome-Wide Identification of Genes Involved in Acid Stress Resistance of *Salmonella* Derby

**DOI:** 10.3390/genes12040476

**Published:** 2021-03-25

**Authors:** Dan Gu, Han Xue, Xiaohui Yuan, Jinyan Yu, Xiaomeng Xu, Yu Huang, Mingzhu Li, Xianyue Zhai, Zhiming Pan, Yunzeng Zhang, Xinan Jiao

**Affiliations:** 1Jiangsu Co-Innovation Center for Prevention and Control of Important Animal Infectious Diseases and Zoonoses, Yangzhou University, Yangzhou 225009, China; 006491@yzu.edu.cn (D.G.); xhxxdjy@163.com (H.X.); xiaohui_yuan@outlook.com (X.Y.); eve_421@163.com (J.Y.); xiao1997710@163.com (X.X.); huangyu20212021@163.com (Y.H.); lmzdyx1997@163.com (M.L.); DX120180123@yzu.edu.cn (X.Z.); zmpan@yzu.edu.cn (Z.P.); jiao@yzu.edu.cn (X.J.); 2Jiangsu Key Laboratory of Zoonosis, Yangzhou University, Yangzhou 225009, China; 3Joint International Research Laboratory of Agriculture and Agri-Product Safety of the Ministry of Education, Yangzhou University, Yangzhou 225009, China; 4Key Laboratory of Prevention and Control of Biological Hazard Factors (Animal Origin) for Agrifood Safety and Quality, Ministry of Agriculture of China, Yangzhou University, Yangzhou 225009, China

**Keywords:** *Salmonella* Derby, Tn-seq, acid resistance, CpxAR, CasC/CasE

## Abstract

Resistance to and survival under acidic conditions are critical for *Salmonella* to infect the host. As one of the most prevalent serotypes identified in pigs and humans, how *S*. Derby overcomes acid stress remains unclear. Here, we de novo sequenced the genome of a representative *S.* Derby strain 14T from our *S.* Derby strain stock and identified its acid resistance-associated genes using Tn-seq analysis. A total of 35 genes, including those belonging to two-component systems (TCS) (*cpxAR*), the CRISPR-Cas system (*casCE*), and other systems, were identified as essential for 14T to survive under acid stress. The results demonstrated that the growth curve and survival ability of Δ*cpxA* and Δ*cpxR* were decreased under acid stress, and the adhesion and invasion abilities to the mouse colon cancer epithelial cells (MC38) of Δ*cpxR* were also decreased compared with the wild type strain, suggesting that the TCS CpxAR plays an essential role in the acid resistance and virulence of *S*. Derby. Also, CasC and CasE were found to be responsible for acid resistance in *S*. Derby. Our results indicate that acid stress induces multiple genes’ expression to mediate the acid resistance of *S*. Derby and enhance its pathogenesis during an infection.

## 1. Introduction

*Salmonella* Derby is one of the most prevalent *Salmonella* serotypes worldwide, and can be isolated from pigs, poultry, and humans, particularly children and the elderly [1,2,3]. In Europe, *S*. Derby is the 5th most frequently identified serotypes in humans [4]. In France, the *S*. Derby strains isolated from pork and poultry were spread across four different genetic lineages corresponding to four multi-locus sequence types (MLST): ST39, ST40, ST71, and ST682 [5]. The *S*. Derby strains isolated from pork samples were mainly affiliated with ST40, and the ST40 strains was responsible for 94% of human infections [2]. In China, *S*. Derby was also identified as one of the common serotypes in pigs and humans, with the strains divided into two MLSTs: ST71 and ST40 [6]. Molecular epidemiology analysis showed that *S*. Derby could be transferred along the pork production chain, and further potentially transferred to humans [6]. The *Salmonella* pathogenicity island (SPI) -1, -2, -3, -4, and -5 regions were detected in the *S*. Derby isolates, suggesting the potential pathogenicity of *S*. Derby [5]. The majority of the *S*. Derby ST40 isolates contained all SPI-23 genes, which have been reported as the critical virulence factor for adhesion and invasion of *Salmonella* [7,8].

*Salmonella* infection is mainly caused by consuming contaminated water or food. *Salmonella* needs to overcome various stress conditions during infection, such as low pH, high osmotic pressure, low oxygen concentrations, and antimicrobial peptides [9]. Furthermore, previous studies have shown that the intragastric environment’s low pH plays an essential role in resisting *Salmonella* infection [10,11,12]. Elderly patients are more likely to suffer infection with *Salmonella*, which is probably related to reducing in gastric acid secretion after 60 [13,14]. After infecting specific pathogen free (SPF) pigs, *S*. Derby could colonize the tonsils, mesenteric lymph nodes, ileum, and cecum, and the highest level contamination of *S*. Derby was found in the tonsils, cecum, and jejunum [15]. Therefore, *S*. Derby is exposed to the stomach and intestinal acids during infection, and examining the fitness of *S*. derby under the acid stress condition is important for understanding its virulence and pathogenicity.

*Salmonella* is an intracellular pathogen that can invade the small intestine’s epithelium cells and survive in an acidified macrophage vacuole with an external pH of 4–5 [16,17,18]. *Salmonella* has evolved a variety of mechanisms to adapt to a low pH environment. Several genes are responsible for tolerating the low pH environment, such as the two-component system (TCS), sigma factors, and those coding for outer membrane proteins. The two-component system PhoP/Q, EnvZ/OmpR could sense the pH and regulate the downstream genes promoting resistance to acid stress [19,20,21]. The alternative sigma factor RpoE could be activated by the acid stress, which is vital for *Salmonella* Typhimurium survival in SCV [22]. The expression and translation of *rpoS* is increased under the acid shock [23]. The inducible lysine decarboxylase and arginine decarboxylase systems play an essential role in the maintaining of intracellular pH in *Salmonella* [24]. Furthermore, the transport protein KdpA, the nitrate reductase subunit NarZ, and the genes involved in NAD^+^/NADH metabolism were significantly upregulated under acid stress [25,26]. Also, the acidic conditions could be a signal to induce the expression of virulence genes, and help *Salmonella*’s intercellular survival [27]. However, our understanding of how *S.* Derby resists acidic stresses to infect the host remains elusive.

Our laboratory has collected more than 400 *S*. Derby isolates from the pork production chain and humans. These strains were divided into ST40 and ST71 by the MLST typing approach [6]. More than 80 strains were then selected to determine the virulence of the *S*. Derby, and 14T-T8N3 (14T-herein), a strain affiliated with ST40, the most prevalent ST type as revealed by our previous epidemiological investigation [6], exhibited the highest virulence. Here, we determined *S*. Derby 14T’s growth ability under different pH conditions, and demonstrated that pH4, similar to the acidic environment in the pig stomach after eating [28,29,30], delayed the growth rate but not the maximum biomass of *S.* Derby 14T. We then generated a high-quality genome assembly of 14T using a de novo long- and short-reads hybrid assembly approach, and used this to identify the essential acid stress genes of *S*. Derby 14T by Tn-seq.

## 2. Materials and Methods

### 2.1. Bacterial Strains, Plasmids, and Cell Lines

The strains and plasmids used in this study are listed in Table 1. *Salmonella* Derby 14T-T8N3 (14T) was isolated from pork sample in 2014, and stored in Luria-Bertani (LB) broth with 25% glycerol at −70 °C by Jiangsu Key Laboratory of Zoonosis. The *S*. Derby T14 wild type strain (WT) and mutant strains were activated by streaking onto LB plates and were cultured at 37 °C for 24 h.

The MC38 (Mouse colon cancer epithelial cell) cell line was purchased from Hunan Fenghui Biotechnology Co., Ltd. (Hunan, China). (catalog no. CL0203). Mouse colon cancer epithelial cells (MC 38) were grown in Dulbecco’s Modified Eagle Medium (DMEM) (Gibco, Grand Island, NY, USA) with 10% fetal bovine serum (FBS) (Gibco, Grand Island, NY, USA) and 1% Penicillin streptomycin at 37 °C with 5% CO_2_.

### 2.2. Genome Sequencing, Assembly and Annotation of the S. Derby 14T

The *S*. Derby 14T genome was sequenced on a PacBio Sequel system and an Illumina Hiseq 4000 platform, and 253,977 long reads (total length 2.06 Gb, N50 9173 bp, max length 45,829 bp) and 11,020,605 ×2 150 bp paired-end short reads were generated by the PacBio and Illumina sequencing, respectively. The short- and long- reads were fed to Unicycler hybrid assembler v0.4.7 for de novo assembly [34]. The genome was annotated using PROKKA v1.14.5 [35]. The COG annotation was assigned to the genes using eggNOG [36]. The genome was visualized using the CGView server (http://stothard.afns.ualberta.ca/cgview_server/, accessed on 10 January 2021) [37].

### 2.3. Transposon Insertion Library Construction

A Tn5 transposon plasmid pKWM2, which contains a gene for resistance to kanamycin, was used to construct the transposon mutant library. The donor strain *E. coli* WM3064 which was used to deliver pKWM2 in this study is an auxotrophic strain whose growth relies on the supplementation of diaminopimelic acid (DAP) in the medium [31]. The *S*. Derby 14T and pKWM2-carrying *E. coli* WM3064 were cultured in LB and LB supplemented with 300 μM diaminopimelic acid (DAP) for 12 h, respectively, and then diluted into new corresponding media and cultured to OD_600_ of 0.4–0.6. The mixtures of *S*. Derby and WM3064 were spotted into 0.22 μm hydrophilic membranes overlaid on LB Agar plates supplemented with 300 μM DAP, and cultured at 37 °C for 12 h. The bacteria in the mixtures were washed by the PBS from the hydrophilic membranes and coating into the LB plates with 50 μg/mL kanamycin and 12.5 μg/mL tetracycline. *S*. Derby 14T was resistant to tetracycline. Therefore, with the presence of tetracycline and absence of DAP in the medium, *E. coli* WM3064 did not grow. After culturing for 12 h, all of the colonies were collected and stored in LB broth with 25% (*v*/*v*) glycerol at −70 °C.

### 2.4. Library Construction, Sequencing and Bioinformatics Analysis

The transposon insertion library was cultured in LB medium with pH 7.0 at 37 °C for 12 h, and then diluted into the new LB medium at pH 7.0 or pH 4.0, respectively. After culturing for 12 h, bacteria were collected, and the genomic DNA was extracted by the TIANamp Bacteria DNA kit (Tiangen, Beijing, China). The library construction and sequencing were conducted as previously described [38]. The data were analyzed using TRANSIT ver. 3.1.0 [39]. Briefly, the raw Tn-seq reads were processed using the TPP tool implemented in TRANSIT with parameter protocol Tn5, and the PROKKA annotated 14T genome was used as the reference. The genes with a differential abundance of insertions (Benjamini–Hochberg adjusted *p*-value < 0.05) were identified using the resembling method implemented in TRANSIT with parameter-a used. The essential genes were determined using the Tn5Gaps method with default parameters.

### 2.5. Construction of Gene Deletion Mutant Strains

The deletion mutant strains were constructed via the double exchange of homologous recombination, as previously described [40]. The primers used for the construction of deletion mutant strains are listed in Table 2. The upstream and downstream DNA fragments were amplified using PCR, and inserted into the suicide plasmid pDM4 digested with Sac I and Sal I by a ClonExpress Multis one-step cloning kit (Vazyme, Nanjing, China). The recombined plasmids were transferred into *E. coli* SM10 *λpir* and conjugated into the WT to select the single-cross strains at the LB plates with 12.5 μg/mL tetracycline and 40 μg/mL chloramphenicol. Then the double-cross strains were selected at LB plated with 15% sucrose, and verified by PCR and sequencing.

### 2.6. Quantitative Real Time PCR

*S*. Derby 14T was cultured at pH 7.0 or pH 4.0 at 37 °C for 6 h, and the total RNA was purified by RNeasy Plus Mini Kits (QIAGEN, USA) following the manufacturer’s instructions. NanoDrop determined the quality of total RNA. The genomic DNA was removed using DNase I (Invitrogen, Carlsbad, CA, USA) and cDNA was synthesized using a PrimeScript RT reagent Kit (TaKaRa Biotechnology Co. Ltd., Dalian, China). The primers used for qRT-PCR were listed in Table 2. The qRT-PCR reactions were performed with 10 μL Fsu SYBR Green Marter Rox (Roche, Basel, Switzerland), 2 μL cDNA, 0.6 μL for forward and reverse primers (Table 2), and 6.8 μL RNase free water. The mix reactions were measured with the Applied Biosystems 7500 real-time system (Applied Biosystems, Foster City, CA, USA). *gyrB* was used as a control gene, and the transcription levels of target genes were determined by 2^−∆∆Ct^.

### 2.7. The Growth Characteristics of WT and Deletion Mutant Strains

The *S*. Derby 14T and deletion mutant strains were cultured at 37 °C for 16 h. The cultures were diluted into conical flasks containing 20 mL LB broth medium with different pHs (pH 2.0, pH 4.0, pH 5.5, or pH 7.0), and then incubated at 37 °C with 180 rpm/min. At the indicated time points, the cell densities of each strain were measured at 600 nm. The initial cell density of these strains was adjusted to an OD_600_ of 0.05, and the pH of each medium was adjusted with 3M HCl.

### 2.8. Survival Analysis

The cultures of *S*. Derby 14T and deletion mutant strains were diluted into the new LB medium, and then incubated until they reached an OD_600_ of 1.0. 1 mL of each culture was resuspended in LB medium at pH 7.0 or pH 4.0, and then static cultured at 37 °C. At the indicated time points, an aliquot of each culture was removed and serially diluted with PBS, and then plated onto the LB plates to determine the survival rate.

### 2.9. Adhesion and Invasion Assay of MC 38 Cells

The *S*. Derby 14T WT, Δ*cpxA*, and Δ*cpxR* were cultured in LB medium at 37 °C for 16 h, the cultures diluted into the new LB medium and incubated to an OD_600_ of 1.0. The MC38 cells were seeded into 24 well plates with 4 × 10^5^ cells per well, and overnight cultured at 37 °C with 5% CO_2_. A total of 1 mL of bacteria was collected and washed twice with DMEM, and then added to each well with a MOI of 20:1. The cells were incubated at 37 °C for 1 h. For adhesion, the cultured cells were washed twice with DPBS (Gibco, Grand Island, NY, USA), and lysed with 0.1% Triton X-100. The lysates were serially diluted and the appropriate dilutions were coated on the LB plates to calculate the number of bacteria. For invasion, the cultured cells were washed twice with DPBS, while 100 µg/mL gentamycin was added to kill the extracellular bacteria, and cultured at 37 °C with 5% CO_2_. After further incubation for 1 h, the 0.1% Triton X-100 was used to lyse the cells, and the number of bacteria was calculated.

### 2.10. Data Availability

The genome sequence and Tn-seq data were deposited in the CNGB database under the Bioproject accession no CNP0001625.

## 3. Results

### 3.1. Overview of the S. Derby 14T Genome

The genome sequence of strain 14T was determined using a short- and long- reads-based hybrid de novo assembly approach, and an assembly composed of 11 scaffolds was generated, with a total length of 4,914,080 bp (Figure 1A and Table 3). The two longest scaffolds were 4,073,894 bp and 805,643 bp long, respectively, accounting for 99.3% of the assembly’s total length. Notably, 4579 protein-coding genes, 22 rRNA, and 84 tRNA coding genes were predicted in the 14T genome. The Clusters of Orthologous Genes (COG) annotation was successfully assigned to 95.7% of these genes (Figure 1B).

### 3.2. Identification of Genes Involved in Acid Stress Resistance by Tn-seq Analysis

To evaluate *S*. Derby 14T’s growth ability under low pH, we characterized the growth curves of WT in the LB medium with pH 2.0, pH 4.0, pH 5.5, and pH 7.0. The results demonstrated that the WT strain could not survive at pH 2.0, and the growth curve of WT cultured at pH 5.5 exhibited no significant difference compared to that at pH 7.0. However, the WT cultured in pH 4.0 exhibited a significantly delayed growth rate, but reached a similar maximum biomass to the pH 7.0 condition (Figure 2A). Interestingly, pH 4.0 represents the actual stomach acidity of pigs after eating [28,29,30]. Therefore, we constructed a high-throughput Tn5-based mutant library of *S*. Derby 14T, and determined the genes involved in acid stress resistance by Tn-seq analysis to identify the genes responsible for acid resistance (pH 4.0).

The mutant library was first grown at pH 7.0 in LB medium, and then challenged at pH 4.0. After culturing for 12 h, the genomic DNA of *S*. derby 14T under pH 7.0 and pH 4.0 conditions were extracted, and the sequencing libraries were constructed. In total, 26,988,364 single-end valid reads (i.e., reads with valid Tn prefix, and insert size >20 bp) (on average 4,498,060 reads per sample, *n* = 6) were generated from the six Tn-seq libraries, and 23,672,555 reads (87.7% of the total reads) were successfully mapped back to the 14 T genome as revealed by the TPP tool. Based on the Tn-seq data generated from the three samples grown at pH 7.0, 581 essential genes were identified using the Tn5Gaps method implemented in the TRANSIT tool, and accounted for 12.7% of all 4579 genes (Figure 2B). The majority of the essential genes were mainly enriched in the core cellular metabolic pathways, including translation, ribosomal structure and biogenesis (19.1%), nucleotide transport and metabolism (11.5%), energy production and conversion (8.9%), and cell wall/membrane/envelope biogenesis (8.1%) (Figure 1A and Figure 2C).

Overall, 35 genes were identified as the acid essential genes, including 17 genes with a diminished abundance of insertions in the pH 4.0 condition, and 18 genes with a greater abundance of insertions (Table 4). These genes were located across the genome (Figure 2D). As expected, many acid-stress-associated genes were identified by Tn-seq, such as the two-component system CpxA and PhoP/PhoQ, which have been identified as responsible for acid resistance in *E. coli* and *Salmonella* Typhimurium, respectively [41,42,43]. *RseB*, *lysP,* and *degP* also showed significantly lower insertions in the pH 4.0 condition than at pH 7.0, which have been identified as being involved in the regulated intramembrane proteolysis (RIP) responsible for the environmental stress [44]. Furthermore, the CRISPR system cascade subunits *casC* and *casE*, the transporter encoding genes *kup*, *crcB,* and *clcA,* also showed a diminished abundance of insertions in the pH 4.0 condition. Taken together, these results demonstrated that these genes could be involved in the acid resistance in *S*. Derby 14T.

### 3.3. Validation of the Tn-seq Identified Genes with qRT-PCR

Ten genes identified by the Tn-seq analysis were selected for verification by qRT-PCR. The qRT-PCR results showed that the transcription level of *clcA* was highly up-regulated after stimulation by acidic conditions (Figure 3). Furthermore, the expression levels of *casC*, *casE,* and *mgrB* also increased under the pH 4.0 condition compared to pH 7.0. The other genes showed no significant differences in expression between the pH 4.0 and pH 7.0 conditions (Figure 3), likely because they were not involved in acid stress at the transcriptional level.

### 3.4. ClcA Was Responsible for Acid Stress Resistance

To further confirm the function of *clcA* in the acid resistance of *S*. Derby, the deletion mutant strain of *clcA* was constructed, and the growth curve of the mutant under acid stress was determined. The bacterial growth curves of the WT and Δ*clcA* showed no significant difference under the pH 7.0 condition, whereas the growth rate of the Δ*clcA* mutant strain was much lower than that of WT at pH 4.0 (Figure 4). These results indicated that the *clcA* plays a vital role in *S*. Derby 14T’s growth under acid stress.

### 3.5. CpxAR Was Responsible for the Acid Stress Resistance and Virulence of S. Derby

CpxAR is the two-component system that has been reported as involved in acid stress tolerance [41]. The growth properties of the WT, Δ*cpxA*, and Δ*cpxR* strains were also investigated in this study. When the strains were grown in LB medium with pH 7.0, the growth curves showed no significant difference between the WT and Δ*cpxA* or Δ*cpxR* mutant strains; however, the Δ*cpxA* strain exhibited dramatic growth defect under the stress condition of pH 4.0, while the growth activity of the Δ*cpxR* strain was slightly decreased compared with the WT strain at pH 4.0 (Figure 5A). The survival ability of Δ*cpxA* and Δ*cpxR* strains significantly decreased compared to the WT after treating with pH 4.0 for 1 h, while no difference was observed at 2 h (Figure 5B). These results demonstrated that CpxA and CpxR contribute to the growth ability of *S*. Derby under the stress condition.

To explore the roles of CpxAR in the virulence of *S*. Derby, the Δ*cpxR* and Δ*cpxR* mutants were inoculated to MC38 cells and the adhesion and invasion abilities were determined. As shown in Figure 5C,D, the adhesion and invasion rates of the Δ*cpxR* strain were significantly decreased compared to the WT, while no difference was observed between Δ*cpxA* and WT, indicating that CpxR contributes to the virulence of *S*. Derby.

### 3.6. CasC/CasE Were Involved in the Acid Stress Resistance

We also constructed the deletion mutant strains of *casC* and *casE* genes, and the growth curves of WT, Δ*casC*, and Δ*casE* strains showed no significant difference under the acid condition (Figure 6A). The survival ability of Δ*casC* was 35.3% and 7.9% after treated with acid stress (pH 4.0) for 1 h and 2 h, which was lower than that of WT (Figure 6B). The survival ability of Δ*casE* also decreased compared to the WT after treatment at pH 4.0 for 1 h and 2 h (Figure 6B). These results showed that CasC and CasE play an essential role in the survival ability of *S*. Derby.

## 4. Discussion

The stomach’s acidic environment can be considered the first barrier against foodborne pathogens, and several organic acids can be produced in the stomach, including lactic, acetic, propionic, and butyric acid [12]. Recent epidemiological studies have demonstrated that *S*. Derby has become one of the most prevalent *Salmonella* serotypes in pigs and humans [1,2,3]. As a foodborne pathogen, *S*. Derby also needs to resist acid stress in the stomach during the infection process, before it can colonize the ileum and cecum [15]. We noted that the growth ability of *S*. Derby 14T was decreased under the pH 4.0 condition compared to the WT grown in pH 7.0 and pH 5.5 at the log-phase, but was not significantly different at the stable phase of growth (Figure 2A). Our results confirmed that *S*. Derby could survive at pH 4.0, and previous studies also indicated that *Salmonella* could survive in the SCV at a pH of 4.0–5.0 [16,17]. Therefore, we used a genome-wide screen method Tn-seq to explore the fitness of *S*. Derby 14T in the acidic environment. The results showed that 35 genes were identified with a significantly diminished or greater abundance of insertions in the LB medium at pH 4.0 (Table 4), indicating these genes were involved in *S*. Derby 14T’s acid stress response.

The 35 genes identified by Tn-seq were divided into five major functional groups, i.e., TCS, RIP system, CRISPR system, transport protein, and metabolism (Table 4) to obtain a better understanding of the genes involved in the acid stress. TCS can sense changes in the environment and regulate downstream genes’ expression in response to environmental changes [45]. The PhoP/PhoQ TCS could respond to the low Mg^2+^, low pH, and cationic antimicrobial peptides in *E. coli*, *Shigella flexneri*, and *S.* Typhimurium [46,47,48]. The PhoP/PhoQ and its regulator protein MgrB were also identified as involved in the acid stress response of *S*. Derby 14T (Table 4). During the stress conditions, activated σ^E^ was released by RIP, which is a crucial regulator of the extracytoplasmic stress response and virulence [44]. The activity of σ^E^ could be induced by acid stress in *S*. Typhimurium [22], indicating that the RIP system may also play an important role in acid resistance in *S*. Derby 14T. The fluoride ion transporter CrcB and H^+^/Cl^−^ exchange transporter ClcA, could promote the extrusion of H^+^ and maintain the survival of foodborne pathogens under extreme acid stress [49,50]. Our results also showed that the expression of *clcA* was induced at pH 4.0 (Figure 3), and the growth curve of Δ*clcA* was decreased compared to the WT at pH 4.0 (Figure 4). Our qRT-PCR results also showed that the *casC*, *casE*, *clcA*, and *mgrB* were significantly increased at pH 4.0 (Figure 3), indicating that acid stress could induce these genes’ expression.

The function of TCS CpxAR in acid stress response and virulence was investigated in *S*. Derby 14T. The results showed that the growth curves of Δ*cpxA* mutant strain exhibited growth defects compared to WT under the stress condition of pH 4.0 (Figure 5A). Furthermore, Δ*cpxA* and Δ*cpxR* strains’ survival ability were significantly decreased compared to the WT after treatment at pH 4.0 (Figure 5B). The adhesion and invasion rate of the Δ*cpxR* strain was significantly decreased compared to the WT (Figure 5C,D). Previous studies also showed that TCS CpxAR could directly sense acidic conditions to regulate the expression of *fabAB* in *E. coli* [41], and contributes to stress resistance and virulence in *Actinobacillus pleuropeumoniae* [51]. Moreover, the CpxAR-mediated envelope stress response plays a crucial role in the gut infection of *S*. Typhimurium [52]. Different growth rate and adhesion and invasion abilities of the Δ*cpxR* and Δ*cpxA* strains were observed in this study, these results suggested that other kinase(s) might interplay with the CpxAR TCS in *S. Derby* 14T [53,54,55]. These results indicated that CpxAR plays a vital role in the acid stress response and virulence of *S*. Derby 14T, even though the expression level of *cpxA* and *cpxR* showed no significant difference from WT cultured at pH 4.0 compared to pH 7.0 (Figure 3). This result may be due to the function of TCS being dependent on the transmission of the phosphate group. Thus, we speculate that CpxAR regulation of the acid stress response may depend on the post-translational, not transcriptional level in *S*. Derby 14T.

The expression of *casC* and *casE* was significantly upregulated under the low pH condition (Figure 3), and the survival ability of Δ*casC* and Δ*casE* was decreased compared to the WT at pH 4.0 (Figure 6B). These results suggest that CasC and CasE may participate in the regulation of acid stress of S. Derby. The *casC* and *casE* genes belonged to the type I-E CRISPR-Cas system, and previous studies of this system focused on virulence. A study in *S*. Enteritidis showed that the type I-E CRISPR-Cas system could regulate the expression of biofilm-forming-related genes and the type three secretion system (T3SS), and the invasion capacity of Δ*cas3* to the host cells was also reduced [56]. The ability of *Salmonella* resistance to the low pH environment was beneficial to the survival of *Salmonella* in the cells [18]. Thus, we can assume that the CasC and CasE proteins’ regulation of acid stress may contribute to the host cell infection of *S*. Derby.

## 5. Conclusions

In summary, the whole genome sequence of *S*. Derby 14T was determined, and 581 genes of the total 4579 genes were identified as the essential genes. Moreover, we have identified 35 genes that could potentially affect the survival ability of *S*. Derby in the low pH condition (pH 4.0). We have confirmed that TCS CpxA/R, ClcA, and CasC/CasE play an essential role in the resistance to acid stress. Nevertheless, functions of these genes in virulence of *S*. Derby need to be explored further.

## Figures and Tables

**Figure 1 genes-12-00476-f001:**
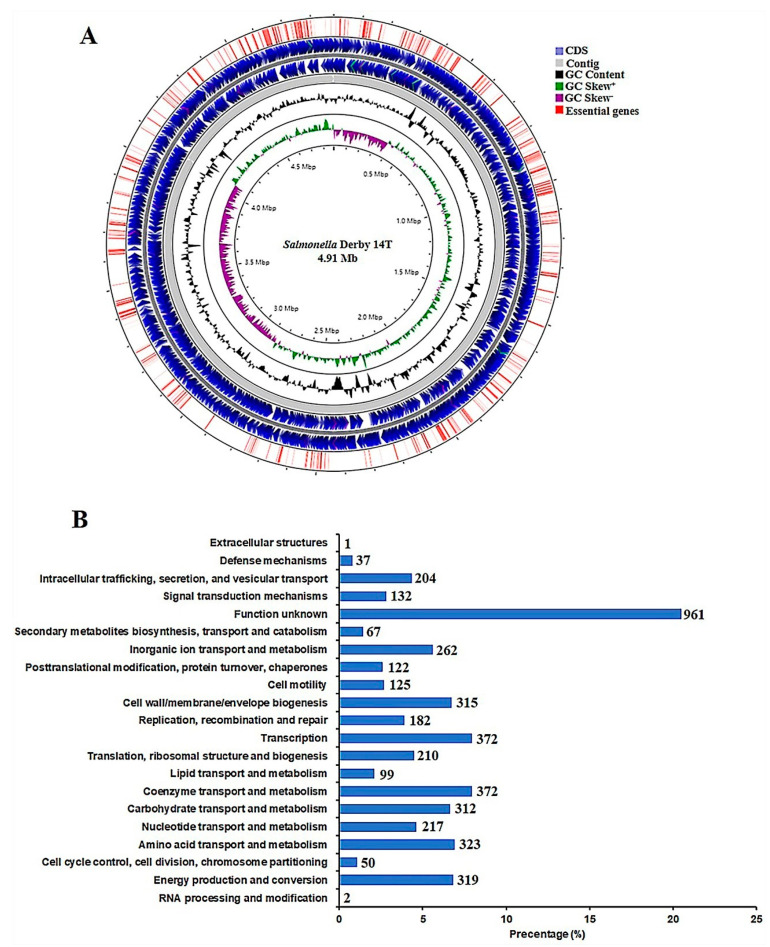
Overview of the *Salmonella* Derby 14T genome. (**A**) Circular representation of the *S*. Derby 14T genome. The contents of the feature rings (starting with the innermost ring) are as follows: GC skew, GC content, CDS on the two strands, and the distribution of 581 conditional essential genes. (**B**) COG annotation of genes in the genome of *S*. Derby 14T. The y-coordinate denotes the COG terms, and the x-coordinate is the percentage of genes affiliated with the term in the total annotated genes.

**Figure 2 genes-12-00476-f002:**
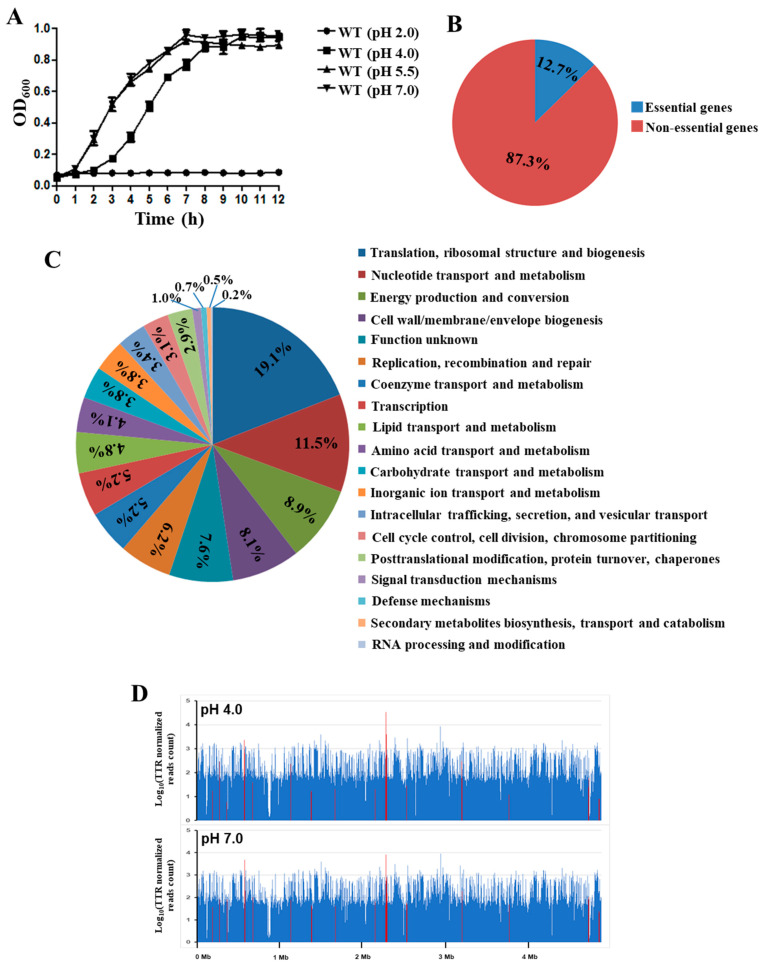
Tn-seq analysis to identify genes involved in acid resistance. (**A**) The growth curves of *S*. Derby 14T at different pH conditions. The WT was grown in LB medium at pH 2.0, 4.0, 5.5 and 7.0, and the OD_600_ values were determined at the indicated time points. The experiments were repeated three times. (**B**) The pie chart of essential genes and non-essential genes as revealed by Tn-seq. (**C**) Functional analysis of essential genes in LB medium at pH 7.0. (**D**) Distribution of the numbers of insertions in the CDS regions across the 14T genome. The upper panel showing the average insertion numbers for each gene of the three replications grown under the pH 7.0 conditions, while the lower panel was those for pH 4.0 acidic conditions. The insertion numbers were shown as Log10 (TTR normalized reads count). Red lines indicate the genes with significantly different insertions, as revealed by the Tn-seq analysis (BH adjusted *p*-value < 0.05).

**Figure 3 genes-12-00476-f003:**
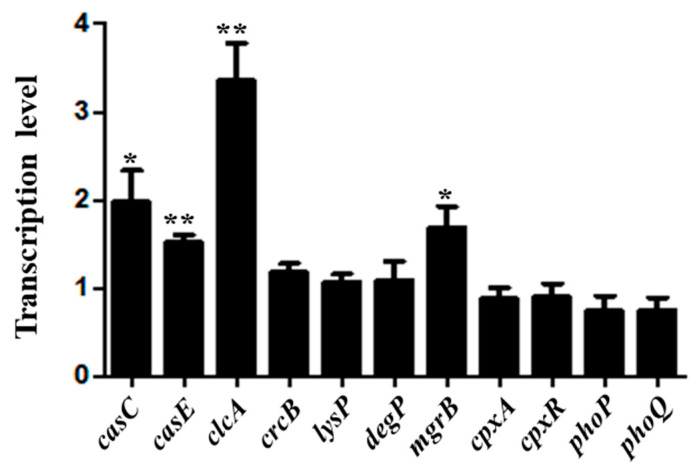
Validation of the genes identified by Tn-seq using qRT-PCR. The WT strain was cultivated at pH 7.0 and pH 4.0, the transcriptional levels of indicated genes show the difference relative to the WT cultured at pH 7.0. *gyrB* was used as the standard gene. Error bars were shown as Standard error of means (SEM)(*n* = 3). *, *p* ≤ 0.05; **, *p* ≤ 0.01 (Student’s *t*-test).

**Figure 4 genes-12-00476-f004:**
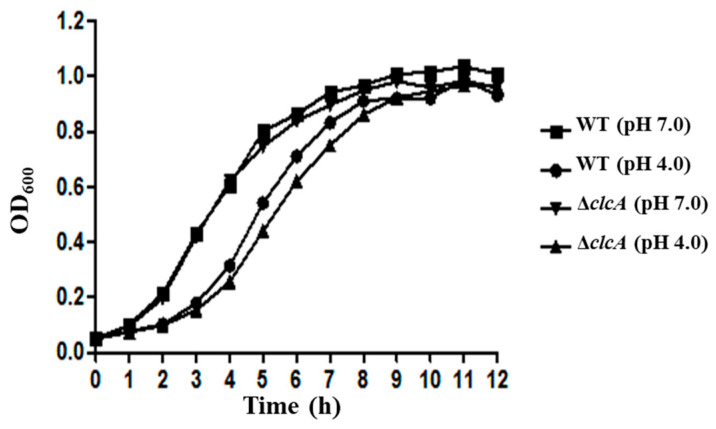
Growth curves of WT and Δ*clcA* in LB medium at pH 4.0 and pH 7.0. The overnight cultures of WT or Δ*clcA* were diluted to a new LB medium at pH 4.0 and pH 7.0, respectively. Then, the bacteria were cultured at 37 °C with 180 rpm, the OD_600_ values were detected at the indicated time points. The experiments were repeated three times.

**Figure 5 genes-12-00476-f005:**
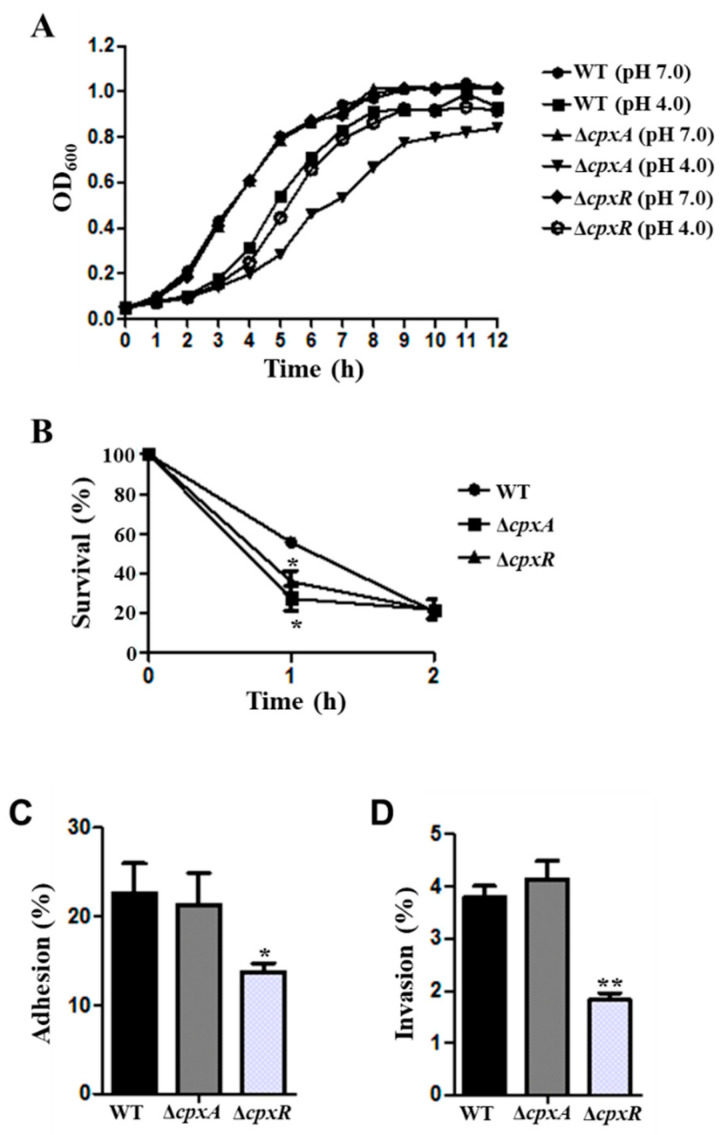
The function of CpxAR in acid resistance and virulence of *S*. Derby. (**A**) Growth curves of WT, Δ*cpxA* and Δ*cpxR* in LB mediums at pH 4.0 and pH 7.0. (**B**) Survival ability of WT, Δ*cpxA* and Δ*cpxR* at pH 4.0. Adhesion (**C**) and invasion (**D**) rate of WT, Δ*cpxA* and Δ*cpxR* after infection with the MC38 cells. All the experiments were repeated three times. *, *p* ≤ 0.05; **, *p* ≤ 0.01 (Student’s *t*-test).

**Figure 6 genes-12-00476-f006:**
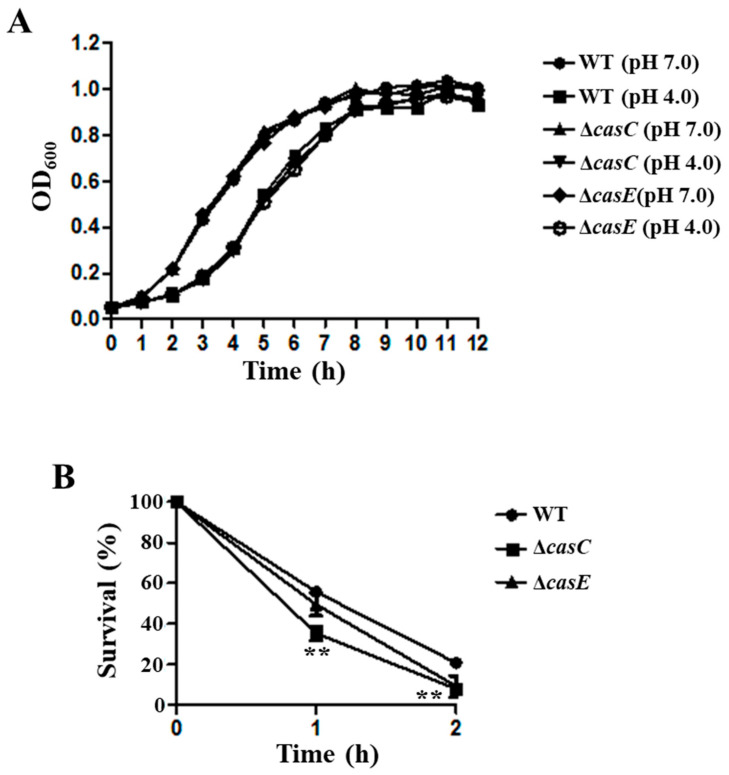
CasC and CasE were involved in the acid resistance of *S*. Derby. (**A**) Growth curves of WT, Δ*casC*, and Δ*casE* in LB medium at pH 4.0 and pH 7.0. (**B**) Survival ability of WT, Δ*casC*, and Δ*casE* at pH 4.0. All the experiments were repeated three times. **, *p* ≤ 0.01 (Student’s *t*-test).

**Table 1 genes-12-00476-t001:** Bacterial strains and plasmids used in this study.

Strain or Plasmid	Relevant Characteristics	Reference
*Escherichia coli*		
WM3064	conjugal donor	[31]
SM10 *λpir*	Host for π requiring plasmids, conjugal donor	[32]
S. Derby		
*S*. Derby 14T	Isolated from pork sample. Tcr	Laboratory collection
Δ*cpxA*	14T-T8N3, in-frame deletion in *cpxA*, Tcr	This study
Δ*cpxR*	14T-T8N3, in-frame deletion in *cpxR*, Tcr	This study
Δ*clcA*	14T-T8N3, in-frame deletion in *clcA*, Tcr	This study
Δ*casC*	14T-T8N3, in-frame deletion in *casC*, Tcr	This study
Δ*casE*	14T-T8N3, in-frame deletion in *casE*, Tcr	This study
Plasmids		
pKWM2	Tn5 transposon vector library, OriT, R6K, Kanr	[31]
pDM4	Suicide vector, pir dependent, R6K, SacBR, Cmr	[33]

**Table 2 genes-12-00476-t002:** Primers used in this study.

Primer Name	Primer Sequence (5′ to 3′)	Target
cpxA-up-F	GAGCGGATAACAATTTGTGGAATCCCGGGAAACATTTAAGTCAGGAAGTGCTGGG	For *cpxA* deletion mutant
cpxA-up-R	TTCGACAATCGGATCGTTCGCAAGTTCAGCTTCTA	For *cpxA* deletion mutant
cpxA-down-F	CGAACGATCCGATTGTCGAAAGCGCCATGCAGCAG	For *cpxA* deletion mutant
cpxA-down-R	AGCGGAGTGTATATCAAGCTTATCGATACCATCTCTTGAGGAGCTTTGGGAGCGG	For *cpxA* deletion mutant
cpxA-out-F	GCTCCTCGAAATGGAAGGTTTTAAT	For *cpxA* deletion mutant
cpxA-out-R	GCTCGCTACAAGTGGGTGAAGAAGG	For *cpxA* deletion mutant
cpxR-up-F	GAGCGGATAACAATTTGTGGAATCCCGGGAAAAAAACTGAATGCCAGCGTTGAGG	For *cpxR* deletion mutant
cpxR-up-R	GTTAGAAATATCGATGCTGTCATCCAAAAGCTCAA	For *cpxR* deletion mutant
cpxR-down-F	ACAGCATCGATATTTCTAACCTGCGCCGCAAACTG	For *cpxR* deletion mutant
cpxR-down-R	AGCGGAGTGTATATCAAGCTTATCGATACCCGTCCTTCAGAGGTCACCAGTAATA	For *cpxR* deletion mutant
cpxR-out-F	TAACCAGCCGTCCATAGGTTTGATT	For *cpxR* deletion mutant
cpxR-out-R	CGACATCACCAGCAGGTCATTGATC	For *cpxR* deletion mutant
clcA-up-F	GAGCGGATAACAATTTGTGGAATCCCGGGAATATGCTTTGCCATCGACATCGTAC	For *clcA* deletion mutant
clcA-up-R	CTGGTAGTTAATTAACCGGCGAATCTGATCTCTGC	For *clcA* deletion mutant
clcA-down-F	GCCGGTTAATTAACTACCAGCTCATTTTGCCAATG	For *clcA* deletion mutant
clcA-down-R	AGCGGAGTGTATATCAAGCTTATCGATACCCCTTCGTTAACCTGATCGTCAAAGG	For *clcA* deletion mutant
clcA-out-F	AGACAGTCTGCGTGGCCGTGGTAGC	For *clcA* deletion mutant
clcA-out-R	TACGCTTATCGGGCCTGGAACATCT	For *clcA* deletion mutant
casC-up-F	GAGCGGATAACAATTTGTGGAATCCCGGGATGATAAGCAGCAAAAATTCGCCGCA	For *casC* deletion mutant
casC-up-R	CATTCATATTCAGGCTCTGAGAGGAGATACGCAGA	For *casC* deletion mutant
casC-down-F	TCAGAGCCTGAATATGAATGAGGTCTATGCACAGG	For *casC* deletion mutant
casC-down-R	AGCGGAGTGTATATCAAGCTTATCGATACCGCACAGCAAAAATTGGTAATGACGA	For *casC* deletion mutant
casC-out-F	CTGCTTGTCTGAGGGATTTCGCTCC	For *casC* deletion mutant
casC-out-R	AGCGGCAACGCCAGAGGGTGACTTT	For *casC* deletion mutant
casE-up-F	GAGCGGATAACAATTTGTGGAATCCCGGGACTTACACGTTGGCGCAGCTTCAGAC	For *casE* deletion mutant
casE-up-R	GCCAGACGTTTGCTGCCGGGAAAGAGATCCCATAA	For *casE* deletion mutant
casE-down-F	CCCGGCAGCAAACGTCTGGCGCAGGGGTACGGTAA	For *casE* deletion mutant
casE-down-R	AGCGGAGTGTATATCAAGCTTATCGATACCAGGAATAGACCCGCACTCCCGCCTC	For *casE* deletion mutant
casE-out-F	GGGATTATCACACGGTGCAGATGCC	For *casE* deletion mutant
casE-out-R	CGGCGATCTCAAATGCTTTGGGCAC	For *casE* deletion mutant
gyrB-q-F	TGATTGCGGTGGTTTCCGTA	qRT-PCR
gyrB-q-R	GACGACGATTTTCGCGTCAG	qRT-PCR
cpxA-q-F	AAGCTGAACTTGCGAACGAT	qRT-PCR
cpxA-q-R	CATCTCTACGCGGCCATATT	qRT-PCR
cpxR-q-F	TTGATGATGACCGAGAGCTG	qRT-PCR
cpxR-q-R	TACCGTTTTTCTTCGGCATC	qRT-PCR
clcA-q-F	CTCAGCAAATTGTGCGCTTA	qRT-PCR
clcA-q-R	CCAGTAACGCCGAAAGGATA	qRT-PCR
lysP-q-F	ACAACTGGGCGGTGACTATC	qRT-PCR
lysP-q-R	TACGCCAACGATGATGAAGA	qRT-PCR
phoQ-q-F	CTCGCCAAATGGGAAAATAA	qRT-PCR
phoQ-q-R	CATTCCGGTTGAATGCTTTT	qRT-PCR
phoP-q-F	TGCGCGTACTGGTTGTAGAG	qRT-PCR
phoP-q-R	TCATCCGGCAGACCTAAATC	qRT-PCR
degP-q-F	GTATGCCGCGTAATTTCCAG	qRT-PCR
degP-q-R	GAATTTACGCCCATCGCTAA	qRT-PCR
casE-q-F	TTATCGCCGCGAAGAGTTAC	qRT-PCR
casE-q-R	GGCTATCACCCTGCGTTTTA	qRT-PCR
mgrB-q-F	AATTTCGATGGGTCGTTCTC	qRT-PCR
mgrB-q-R	CGCAAATACCGCTGAAAAAT	qRT-PCR
crcB-q-F	CCCGGTGGATGCTAAGTATG	qRT-PCR
crcB-q-R	GGTCGTAATGAGCACTTTCCA	qRT-PCR
casC-q-F	AGAACATCGCCAACTGCTTT	qRT-PCR
casC-q-R	CAGTTCCGCTTCCTCTTCTG	qRT-PCR

**Table 3 genes-12-00476-t003:** Features of the *S*. Derby 14T genome.

Features	Value
Total reading base pairs (bp)	4,914,080
Scaffold number	11
N50 (bp)	4,073,894
GC content (%)	52.03
tRNAs	84
rRNAs	22
Protein-coding sequences	4579

**Table 4 genes-12-00476-t004:** Genes showing significantly different insertions under the acid condition identified by Tn-seq.

Gene	Description	log2FC
T141_00187	Sensor histidine kinase, CpxA	−1.08
T141_02433	Virulence transcriptional regulatory protein, PhoP	−1.07
T141_02432	Virulence sensor histidine kinase, PhoQ	−0.99
T141_03065	PhoP/PhoQ regulator, MgrB	−1.72
T141_01611	Sigma-E factor regulatory protein, RseB	−1.01
T141_02070	Lysine-specific permease, LysP	−1.73
T142_00594	Periplasmic serine endoprotease, DegP	−0.86
T141_01333	CRISPR system Cascade subunit, CasC	−1.41
T141_01335	CRISPR system Cascade subunit, CasE	−1.65
T141_00366	Low affinity potassium transport system protein, kup	−3.8
T141_03609	Putative fluoride ion transporter, CrcB	−2.03
T142_00588	H(+)/Cl(-) exchange transporter, ClcA	−1.7
T142_00709	Glutamate 5-kinase, ProB	−1.67
T141_00659	Endoglucanase, BcsZ	−1.42
T142_00710	γ-glutamyl phosphate reductase, ProA	−1.38
T141_01331	hypothetical protein	−1.15
T141_00565	hypothetical protein	−1.06
T141_02199	D-inositol-3-phosphate glycosyltransferase, MshA	1.09
T141_02191	dTDP-4-dehydrorhamnose 3%2C5-epimerase, RfbC	1.74
T141_02189	dTDP-4-dehydrorhamnose reductase, RfbD	1.98
T141_02193	Glucose-1-phosphate cytidylyltransferase, RfbF	2.29
T141_02194	CDP-glucose 4%2C6-dehydratase, RfbG	2.02
T141_02196	CDP-abequose synthase, RfbJ	2.64
T141_02201	Mannose-1-phosphate guanylyltransferase, RfbM	1.23
T141_02198	Abequosyltransferase, RfbV	2.37
T141_01096	Cell division protein, FtsP	1.54
T141_00272	Sec-independent protein translocase protein, TatC	1.81
T141_02202	Phosphoglucosamine mutase, GlmM	1.9
T141_02192	CDP-6-deoxy-L-threo-D-glycero-4-hexulose-3-dehydrase reductase, AscD	2.05
T141_00569	D-inositol-3-phosphate glycosyltransferase, MshA	2.09
T141_02195	dTDP-4-dehydro-2%2C6-dideoxy-D-glucose 3-dehydratase, SpnO	2.12
T141_03062	Peptidoglycan D%2CD-transpeptidase, FtsI	2.41
T141_02203	hypothetical protein	1.58
T141_00570	hypothetical protein	1.7
T141_02200	hypothetical protein	1.83

## Data Availability

The genome sequence and Tn-seq data were deposited in the CNGB database under the Bioproject accession no. CNP0001625.

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
