# Peer review of "Genome-Wide Identification of Genes Involved in Acid Stress Resistance of Salmonella Derby"

_genes, 2021, doi:10.3390/genes12040476_

Round 1

Reviewer 1 Report

Authors describe sequencing of a transposon insertion library of a representative S. Derby strain grown under different pH conditions to identify genes involved in acid resistance of S. Derby. They use the sequencing analysis information to select genes to generate mutants which were then used to study the growth curves, survival, adhesion and invasion of the mutant strains compared to the WT. The paper is well written and easy to follow.

Please check correct references are cited in the introduction (reference 4, lines 42 - 44).

Figure 5A. Difficult to read the figure. The text describes slower growth (defect) in ΔcpxA and ΔcpxR strains under the stress condition of pH 4.0 however the difference between the WT and ΔcpxR  is negligable, almost non-existent. Cold the authors provide more information on the growth curves ofΔcpxA and ΔcpxR?

If there is no difference in the growth rate of ΔcpxR muyant, the following line also needs to be changed accordingly "These results suggested that CpxAR plays an essential role in the acid stress tolerance and virulence of S. Derby" (lines 296 and 297) and the discussion thereafter.
  . 

Author Response

Please check correct references are cited in the introduction (reference 4, lines 42 - 44).

A: Thanks. The reference should be ref. 5. This reference has been corrected in the revised manuscript.

Figure 5A. Difficult to read the figure. The text describes slower growth (defect) in ΔcpxA and ΔcpxR strains under the stress condition of pH 4.0 however the difference between the WT and ΔcpxR is negligable, almost non-existent. Cold the authors provide more information on the growth curves ofΔcpxA and ΔcpxR?

A: Thanks for this valuable suggestion. We revised the sentence as below “

When the strains were grown in LB medium with pH 7.0, the growth curves showed no significant difference between the WT and ΔcpxA or ΔcpxR mutant strains; however, the ΔcpxA strain exhibited dramatic growth defect under the stress condition of pH 4.0, while the growth activity of the ΔcpxR strain was slightly decreased compared with the WT strain at pH 4.0 (Figure 5A). “

If there is no difference in the growth rate of ΔcpxR muyant, the following line also needs to be changed accordingly "These results suggested that CpxAR plays an essential role in the acid stress tolerance and virulence of S. Derby" (lines 296 and 297) and the discussion thereafter.

A: Thanks for this valuable suggestion. We deleted this sentence in the main text, and added discussion as below “Different growth rate and adhesion and invasion abilities of the ΔcpxR and ΔcpxA strains were observed in this study, these results suggested that other kinase(s) might interplay the CpxAR TCS in S. Derby 14T [53-55].

Attached please find our revised manuscript with and without tracks. Thanks.

Reviewer 2 Report

Authors elucidated the role of 35 genes in the acid stress resistance in salmonella Derby. The role of some genes has already been elucidated in other Salmonella serotypes, but this is the first study on the serotype Derby. The study is well done and complete. Genes involved in acid stress résistance have been identified using transposon insertion library, mutants generation and adhesion and invasion of mouse colon cancer epithelial cells. Experiments are well conducted, and results clearly exposed. While the role of some genes like cpx, cas, and crc in acid stress response is known for other Salmonella serotypes, the study added more genes (a total of 35) to this field.

Minor comment:

Methods. 2.3. Transposon insertion library construction: this step was accomplished by mixing a receptor strain (S. Derby) and a donor one (E. coli harboring kanamycin plasmid), instead of introducing only the plasmid into Salmonella. Authors should complete by explaining how Salmonella strains have been selected and separated from E. coli cells.

Author Response

Methods. 2.3. Transposon insertion library construction: this step was accomplished by mixing a receptor strain (S. Derby) and a donor one (E. coli harboring kanamycin plasmid), instead of introducing only the plasmid into Salmonella. Authors should complete by explaining how Salmonella strains have been selected and separated from E. coli cells.

A: Thanks for your valuable suggestion. We revised the methods 2.3 as below

“A Tn5 transposon plasmid pKWM2, which contains a gene for resistance to kanamy-cin, was used to construct the transposon mutant library. The donor strain E. coli WM3064 which was used to deliver pKWM2 in this study is an auxotrophic strain whose growth relies on the supplementation of diaminopimelic acid (DAP) in the medium [31]. The S. Derby 14T and pKWM2-carrying E. coli WM3064 were cultured in LB and LB supple-mented with 300 μM diaminopimelic acid (DAP) for 12 h, respectively, and then diluted into new corresponding media and cultured to OD600 of 0.4-0.6. The mixtures of S. Derby and WM3064 were spotted into 0.22 μm hydrophilic membranes overlaid on LB Agar plates supplemented with 300 μM DAP, and cultured at 37°C for 12 h. The bacteria in the mixtures were washed by the PBS from the hydrophilic membranes and coating into the LB plates with 50 μg/mL kanamycin and 12.5 μg/mL tetracycline. S. Derby 14T was re-sistant to tetracycline. Therefore, with the presence of tetracycline and absence of DAP in the medium, E. coli WM3064 did not grow. After culturing for 12 h, all of the colonies were collected and stored in LB broth with 25% (v/v) glycerol at -70°C.”

Attached please find our revised manuscript. Thanks.
